# Activin A Limits VEGF-Induced Permeability via VE-PTP

**DOI:** 10.3390/ijms24108698

**Published:** 2023-05-12

**Authors:** Basma Baccouche, Lina Lietuvninkas, Andrius Kazlauskas

**Affiliations:** 1Department of Ophthalmology & Visual Sciences, University of Illinois at Chicago, Chicago, IL 60612, USA; 2Department of Physiology and Biophysics, University of Illinois at Chicago, Chicago, IL 60612, USA

**Keywords:** activin, VEGF, permeability, crosstalk, follistatin, VE-PTP

## Abstract

The clinical success of neutralizing vascular endothelial growth factor (VEGF) has unequivocally identified VEGF as a driver of retinal edema that underlies a variety of blinding conditions. VEGF is not the only input that is received and integrated by the endothelium. For instance, the permeability of blood vessels is also regulated by the large and ubiquitously expressed transforming growth factor beta (TGF-β) family. In this project, we tested the hypothesis that members of the TGF-β family influence the VEGF-mediated control of the endothelial cell barrier. To this end, we compared the effect of bone morphogenetic protein-9 (BMP-9), TGF-β1, and activin A on the VEGF-driven permeability of primary human retinal endothelial cells. While BMP-9 and TGF-β1 had no effect on VEGF-induced permeability, activin A limited the extent to which VEGF relaxed the barrier. This activin A effect was associated with the reduced activation of VEGFR2 and its downstream effectors and an increased expression of vascular endothelial tyrosine phosphatase (VE-PTP). Attenuating the expression or activity of VE-PTP overcame the effect of activin A. Taken together, these observations indicate that the TGF-β superfamily governed VEGF-mediated responsiveness in a ligand-specific manner. Furthermore, activin A suppressed the responsiveness of cells to VEGF, and the underlying mechanism involved the VE-PTP-mediated dephosphorylation of VEGFR2.

## 1. Introduction

Vascular endothelial growth factor A (VEGF) is required for the formation and function of vasculature [1,2]. Mice lacking only one of the two VEGF alleles are embryonic lethal [3], whereas mice heterozygous for other angiogenic factors (PDGF) [4], or even central governors of metabolism, (insulin) are viable [5]. In addition to its essential contribution to angiogenesis during embryonic development, VEGF has been identified as a major mediator of endothelial cell integrity in adult vasculature. Severe vascular defects, progressive endothelial degeneration, and death result when VEGF is deleted in the endothelial lineage [2]. These studies demonstrate the importance of VEGF in vascular homeostasis.

VEGF is also involved in the pathogenesis of diabetic complications, such as diabetic retinopathy (DR) and diabetic macular edema (DME). As the level of VEGF increases, it causes leakage and angiogenesis of the retinal vessels, which compromises vision. The clinical relevance of VEGF as a therapeutic target has revolutionized the management of blinding diseases [6] and reinforced the importance of VEGF’s role as a regulator of vascular homeostasis.

The microenvironment of the vasculature provides a plethora of information that influences the behavior of the vascular endothelium. Such input includes the composition of the extracellular matrix (ECM) and the presence of both soluble and ECM-bound factors [7,8,9]. Soluble growth factors, including those of the TGF-β superfamily, act on cells within the vasculature to define the microenvironment [8]. Thus, soluble factors govern the behavior of the endothelium both directly (as described above for VEGF) and indirectly by sculpting the vasculature’s microenvironment.

Crosstalk between VEGF and TGF- β pathways is essential for proper patterning of the vasculature [9,10]. Whether such crosstalk occurs at the level of the microenvironment and/or intracellular signaling remains incompletely understood [11]. Such information will enable the improvement of current therapeutic approaches to allow them to manage VEGF-driven pathology.

Herein, we continue to elucidate the crosstalk between the TGF-β family and VEGF. We discovered that activin limited VEGF’s ability to relax the endothelial barrier and that the underlying mechanism involved increased the expression of VE-PTP, which suppressed VEGF-induced signaling.

## 2. Results

### 2.1. The TGF-β Superfamily Governed Permeability in a Ligand-Specific Manner

While it is well-established that the TGF-β superfamily contributes to vascular homeostasis, the underlying mechanism(s) has not been fully resolved. To this end, we considered whether various members of the TGF-β superfamily influenced basal or VEGF-driven paracellular permeability, which we measured via electric cell-substrate impedance sensing (ECSIS)/transendothelial electrical resistance (TEER). Pilot experiments, which were performed to identify the optimal dose and mode of exposure (pre-treatment versus simultaneous addition with VEGF, unpublished observations), indicated that pre-treating primary human retinal endothelial cells (HRECs) for 48 h with either 10 ng/mL (BMP-9 or TGF-β1) or 50 ng/mL (activin A) induced the greatest effect. More specifically, pre-treating cells with BMP9 significantly increased basal permeability, while TGF-β1 and activin A had no effect (Figure 1a–c). Furthermore, VEGF-induced relaxation was attenuated in activin pre-treated cells, whereas BMP-9 and TGF-β1 did not influence the extent of VEGF-induced permeability. 

These results indicate that certain members of the TGF-β superfamily contribute to the control of either the basal or VEGF-stimulated permeability of HRECs. We proceed to further investigate activin’s ability to blunt a cell’s ability to respond to VEGF. 

### 2.2. Activin Did Not Attenuate Cytokine-Induced Permeability

To test if the activin effect described above was unique to VEGF, we stimulated activin pre-treated cells with either interleukin 1 beta (IL1β), tumor necrosis factor-alpha (TNFα), or a combination of both cytokines. Activin had no effect on permeability driven by either cytokine alone, and it modestly enhanced permeability in response to the cytokine combo (Figure 2).

These data demonstrate that the suppressive effect of activin was not common to all agents capable of relaxing the endothelial barrier. Such observations indicated that activin acted on components of the permeability pathway that were unique to VEGF.

### 2.3. Activin Reduced the Level of eNOS

We asked if activin reduced the expression of the effectors that VEGF uses to relax the barrier. While the level of VEGFR2, proto-oncogene tyrosine-protein kinase (Src), phospholipase C gamma (PLC γ), endothelial nitric oxide synthase (eNOS), extracellular signal-regulated kinase (Erk), and serine-threonine protein kinase (Akt) were unaffected by pre-treating cells with activin, the expression of eNOS was significantly reduced in activin pre-treated cells (Appendix A). However, siRNA-mediated suppression of eNOS did not impair VEGF-induced relaxation (Appendix A). These data indicate that while the level of some of the VEGF effectors was reduced by pre-treating cells with activin, this change was unlikely to account for the cells’ attenuated response to VEGF.

### 2.4. Conditioned Medium Experiment

An alternative way that activin could attenuate VEGF-, but not cytokine-induced permeability is by increasing the expression of soluble VEGFR1, which traps and thereby neutralizes VEGF. This possibility is supported by a report that activin stimulates the production of a VEGF trap [12]. To test if actin induced the production of factors that neutralize VEGF, we collected conditioned medium from cells that were treated with vehicle or activin for 48 h and then tested for its ability to block acute VEGF-dependent phosphorylation of VEGFR2 in naïve HRECs (not pre-treated with activin). Such a conditioned medium had no effect on VEGF’s ability to activate/phosphorylate VEGFR2 (Figure 3a). These results indicate that in our experimental setting, activin did not induce the release of soluble factors that neutralize VEGF.

### 2.5. VEGF-Induced Signaling Was Attenuated in Activin Pre-Treated Cells

Having ruled out the VEGF trap possibility, we asked if pre-treating cells with activin suppressed VEGF-dependent signaling events. Indeed, VEGF-induced phosphorylation of VEGFR2 at both Y1175 and Y951 was reduced in activin pre-treated cells (Figure 3b,c).

Similarly, the VEGF-dependent activation of five signaling enzymes downstream of the receptor was also reduced (Figure 4). These results indicate that activin suppressed VEGF-dependent signaling, which is a plausible explanation for how activin attenuated VEGF-mediated permeability. 

### 2.6. The Effect of Activin Was Dependent on VE-PTP

Protein tyrosine phosphatases dephosphorylate activated receptor tyrosine kinases [13], and three of them (VE-PTP, Density-Enhanced Phosphatase-1(DEP-1), and Protein Tyrosine Phosphatase 1B (PTP1b)) dephosphorylate VEGFR2 [14].

Activin pre-treatment resulted in a statistically significant increase in the expression of Protein Tyrosine Phosphatase Receptor Type B (*PTPRB*) mRNA and protein for VE-PTP (Figure 5), which suggested that increased expression of VE-PTP was involved in the effect of activin.

Two experimental approaches supported this possibility. The siRNA-mediated suppression of VE-PTP eliminated the activin effect, i.e., VEGF-induced permeability was no longer attenuated (Figure 6 and Appendix A). Similarly, inhibiting the activity VE-PTP (with AKB-9778) eliminated the effect of activin on VEGFR2 phosphorylation (Figure 7). These data indicate that activin suppressed VEGF-induced permeability by increasing the expression of VE-PTP, which reduced phosphorylation/activation of VEGFR2 and its downstream effectors.

### 2.7. The Effect of Activin on Additional VEGF–Dependent Cellular Responses

The elucidation of the mechanism by which activin suppresses VEGF-induced permeability (reduced activation of VEGFR2) predicts that other VEGF-driven responses will also be affected. EC migration is an essential component of the angiogenic process, which involves the activation of several signaling pathways that converge on cytoskeletal remodeling, resulting in cell extension, contraction, and forward progression [15].

We investigated the effect of activin on the migration of HRECs using a wound-healing assay in which cells migrate into a cell-free zone created by scratching a confluent monolayer. As expected, VEGF promoted migration, and activin suppressed this response (Figure 8A). In contrast, activin had no effect on serum-driven migration (Figure 8B).

These observations reinforce the concept that activin selectively attenuated responsiveness to VEGF. Furthermore, they show that multiple VEGF-driven responses were affected, which is consistent with activin’s mechanism of action—attenuating the VEGF-induced activation of VEGR2.

### 2.8. Follistatin Overcame the Effect of Activin

Having established that responsiveness to VEGF can be modulated by soluble factors within the microenvironment, such as activin, we extended this line of investigation to follistatin (FST), which binds and thereby inhibits the bioactivity of activin in a variety of experimental settings [16]. More specifically, we considered if FST could overcome activin’s influence on the VEGF-driven barrier relaxation. Indeed, we found that this was the case. Activin lost its ability to suppress VEGF-induced permeability in the presence of FST (Figure 9a). The addition of FST alone had no effect on either the basal or VEGF-induced barrier function (Appendix A). We concluded that FST was capable of overcoming the inhibitory effect of activin on VEGF-driven permeability (Figure 9b). These observations reinforce the concept that the responsiveness of cells to VEGF depends on the concentration of VEGF as well as the presence of other soluble factors.

Because FST is expressed ubiquitously [17], we considered if HRECs produced FST, which would inhibit exogenously-added activin and thereby underestimate its bioactivity. Less than 1 ng/mL FST accumulated in the conditioned medium within a 24 h time period (unpublished observations). This amount of FST was 1000× less than the amount of FST that was necessary to antagonize 50 ng/mL activin (Figure 9). Thus, HRECs did not produce enough FST to blunt activin’s effect in our experimental system.

While the level of FST produced by HRECs was small, it may be sufficient to antagonize a correspondingly low level of endogenously produced activin, which we determined to be 0.1 ng/mL in the conditioned medium after 24 h. If this was true, then targeting endogenously-produced FST would enable endogenously-produced activin and suppress VEGF-induced barrier relaxation. However, we found that the siRNA-mediated suppression of FST (mRNA reduced by 90%) or the addition of an anti-FST antibody had no effect on the basal and VEGF-induced permeability (Appendix A). Furthermore, the amount of endogenously-produced activin (0.1 ng/mL) was far below the concentration of activin that suppressed VEGF-induced permeability (50 ng/mL). We conclude that endogenously-produced activin does not detectably affect the ability of HRECs to respond to VEGF in our experimental system.

## 3. Discussion

In this study, we observed that the concentration of VEGF was not the sole determinant of a cell’s response to VEGF. Activin, whose bioactivity was in turn governed by FST, determined the potency of VEGF. Activin suppressed the responsiveness to VEGF by VE-PTP-mediated attenuation of VEGFR2 activation (Figure 10). Thus, the consideration of other agents within the microenvironment of the vasculature is a potential strategy to improve current anti-VEGF-based therapies to govern pathologies arising from aberrant vascular homeostasis.

The endothelium perceives and integrates information arriving from its microenvironment [18,19]. The input from TGF-β family members is processed via SMADs. For instance, TGF-Β1 and activin engage SMAD2 and SMAD3 to promote proper perfusion and function of the inner retinal vasculature [20]. In contrast, BMP9 acts via SMAD1/5 to control vascular development and barrier function [21,22,23].

The unique effect of BMP9 on the basal barrier function was consistent with the distinct SMADs utilized by this TGF-β family member. While TGF-β1 and activin did not affect barrier function in the absence of VEGF, BMP9 relaxed it to nearly the same extent as VEGF did. These findings suggest that SMAD1/5-regulated genes encode proteins that govern vascular homeostasis. Indeed, BMPs induce reorganization of the actin cytoskeleton, and the expression of junctional proteins is regulated by SMAD1/5 [21,24,25].

Despite the correlation mentioned above, our findings indicate that the type of SMADs utilized by a TGF-β family member is not the only determinant of its effect on the VEGF-driven response. While both TGF-β1 and activin activate SMAD2/3, only activin attenuated VEGF-induced permeability. Identifying the accompanying events that enable the suppression of the response to VEGF is a prerequisite for the development of therapeutic approaches that can limit VEGF-induced permeability. The impressive breadth of TGF-β1’s influence in physiology and pathology [11,26,27,28] suggests that identifying such events may not be a trivial undertaking. 

While this report is focused on activin, the behavior of BMP9-treated cells reveals two concepts. First, the pathways by which BMP9 and VEGF govern the barrier are likely to be non-overlapping because VEGF-induced barrier relaxation was unaffected by the BMP9-driven reduction in basal barrier function. If BMP9 engaged the same pathway as VEGF, then VEGF would not have been able to further relax the barrier in the BMP9-treated cells. Second, the extent to which even a saturating dose of VEGF relaxed the barrier was partial. The degree of VEGF-induced relaxation was substantially increased by pre-treating cells with BMP9. This second concept reinforces the central theme of this report that the concentration of VEGF is only one of the variables that control permeability.

The discovery that activin suppresses multiple VEGF-driven responses indicates activin’s potential to treat diseases, such as DR. However, while activin attenuates responsiveness to VEGF, it modestly promotes permeability to the types of cytokines that are elevated in the vitreous of such patients (Figure 3). Furthermore, our RNAseq data reveal that activin alters the expression of thousands of genes (unpublished observations), which increases the likelihood of undesirable side effects of an activin-based therapeutic approach. 

A better approach may be to increase the expression of VE-PTP, which is required for the activin-mediated attenuation of responsiveness to VEGF. 

Such conclusions emerging from our in vitro experimental setting do not align with the beneficial effect of reducing (instead of increasing) the effect of VE-PTP in patients with DR [29]. VE-PTP has multiple substrates that govern vascular homeostasis, and their relative importance appears to be dependent on the context [30]. In vivo, Tie2 is the prominent VE-PTP substrate, whereas, in the in vitro setting, VEGFR2 is the more important VE-PTP substrate because the Tie2 pathway is only feebly active [31,32]. While this feature of the in vitro system is not a faithful model of the in vivo setting, it simplifies interrogation of the VE-PTP/VEGFR2 relationship.

The growing appreciation that dysfunction of the blood–brain barrier and increased permeability of the brain vasculature contributes to neurological diseases, such as dementia and Alzheimer’s disease [33], suggests that anti-VEGF-based therapies will be used to manage a wider spectrum of diseases in the future.

## 4. Materials and Methods

### 4.1. Materials

Human retinal endothelial cells (HRECs) were purchased from Cell Systems (Kirkland, WA, USA). Lonza™ endothelial cell basal medium-2 (EBM-2, CC3156) and Lonza™ SingleQuots™ endothelial cell growth medium-2MV (EGM-2MV, CC4147) for tissue culture were procured from Lonza Bioscience (Verviers, Belgium) and D-(+)-Glucose (G7021) was procured from Sigma-Aldrich (St. Louis, MO, USA). Recombinant follistatin (5836-FS), recombinant activin A (338-AC), recombinant BMP-9 (3209-BP), Human follistatin Quantikine ELISA Kit (DFN00), and human activin A DuoSet ELISA (DY338) were purchased from R&D systems (Minneapolis, MN, USA).

Recombinant human VEGF-165 (100-20), recombinant human IL-1β (200-01B), recombinant human TNF-α (300-01A), and recombinant human TGF-Β1 (100-21) were purchased from PeproTech, Inc. (Cranbury, NJ, USA). Razuprotafib (329509, known as AKB-9778) was purchased from MedKoo Biosciences. ON-TARGETplus Human FST (L-012221-00-0010) siRNA, ON-TARGETplus Human PTPRB (5787) siRNA, ON-TARGETplus Human NOS3 (L-006490-00-0010), and ON-TARGETplus non-targeting Control siRNAs (D-001810-10-05) were purchased from Horizon. 

The RNeasy Plus mini kit (74104) used for RNA isolation was obtained from QIAGEN (Hilden, Germany). The High-Capacity cDNA Reverse Transcription Kit (4368814) used for cDNA synthesis and the Fast SYBR™ Green Master Mix (4385612) used for real-time PCR were purchased from Applied Biosystems (Thermo Fisher Scientific, Waltham, MA, USA). 

The disposable electrode arrays (8W10E+ PC) used in the transendothelial electrical resistance (TEER) assay for measuring cellular permeability were purchased from ECIS CulturewareTM (Applied BioPhysics, Inc., Troy, NY, USA).

Anti–Enos Ser1177 (9571), Anti-PLC Tyr783 (2821), Anti-VEGFR2 (247), and Anti-P-VEGFR2 1175 (2478) were purchased from Cell Signaling, Inc. (Danvers, MA, USA). Anti-p-VEGFR2 951(ab38473) was purchased from Abcam (Boston, MA, USA), Anti-DEP-1 (sc-21761) was purchased from Santa Cruz (Dallas, TX, USA), and anti-PTP1B was obtained from BD Biosciences (San Diego, CA, USA). Anti-VE-PTP Ab was kindly provided by Dietmar Vestweber (Max Planck Institute for Molecular Biomedicine, Münster, Germany). Anti-rabbit (7074S) and anti-mouse (7076S) IgG HRP-linked Ab were purchased from Cell Signaling, Inc. (CST).

Mini-PROTEAN^®^ TGX™ precast gels were purchased from Biorad, ECL substrates were purchased from Thermo Fisher Scientific, and Bovine Serum Albumin was obtained from (A2153-50G, Sigma-Aldrich, St. Louis, MO, USA).

### 4.2. Culture of Human Retinal Endothelial Cells 

Primary HRECs were used to establish an in vitro model of DR. HRECs were cultured in complete Lonza medium (EBM-2 supplemented with the EGM-2MV SingleQuots kit) containing 30 mmol/L D-glucose for at least 10 days; the medium was changed every 24 h. Condition media was collected after 24 or 48 h and stored at −80 °C until analysis.

### 4.3. TEER Measuring Cellular Permeability

Cell permeability was assessed by measuring changes in TEER using an electrical cell-substrate impedance-sensing ZThera instrument (Applied Biophysics, Troy, NY, USA) housed in a standard tissue culture incubator (maintained at 37 °C and 5% CO_2_), as previously described [34,35]. Briefly, approximately 0.6 × 105 HG HRECs (1/8 of a confluent 10 cm dish) were resuspended in a complete Lonza medium containing 30 mM D-glucose and HRECs were seeded in 8-well chamber slides equipped with gold-coated microelectrodes. A total of 24 h after plating, the media was changed and vehicle or activin was added for 48 h. Following the addition of the test agents, TEER was measured continuously and in real-time. The electric current passing through the endothelial monolayers was measured independently in each chamber.

### 4.4. Western Blot Analysis

HG HRECs were treated with different agents (activin, VEGF, AKB) for indicated time periods. Cells were rinsed with ice-cold phosphate-buffered saline (PBS, pH 7.5) and lysed in an electrophoresis sample buffer (10 mM EDTA; 2% sodium dodecyl sulfate; 0.2 M 2-mercaptoethanol; 20% glycerol; 200 mM Tris-HCl, pH 6.8; and 0.2% bromophenol blue) [34].

Proteins were resolved on a 10% and 12% sodium dodecyl sulfate–polyacrylamide gel and subjected to Western blot analysis. The membranes were blocked in TBST (10 mM Tris, pH 7.5, 150 mM NaCl, and 0.05% Tween-20) containing 5% BSA for 1 h at room temperature and then probed overnight at 4 °C with the specific primary antibody. Following the washing steps, the membranes were incubated for 1 h at room temperature with secondary antibodies.

The specific protein bands were visualized by using an ECL detection kit (Thermo Fisher Scientific) quantified by ImageJ 1.53a. The blots were cropped, and full-length uncropped gel images are presented in a Appendix A file.

### 4.5. Wound-Healing Assay

HRECs were suspended in a complete Lonza medium, then seeded at 10,000 cells per well on 24-well culture dishes precoated with 0.1% gelatin solution (Sigma, St. Louis, MO, USA). Cells were allowed to grow to confluence for 48 to 72h. Prior to the experiment, the media was replaced with a starvation medium (EBM-2 + 2.5% FBS + 0.5% ascorbic acid+ 0.4%hydrocortisone), and after 4 h the confluent monolayer was scratched once using a P200 pipette tip. The medium was replaced with the starvation medium containing vehicle, 2 nM VEGF, or 10% serum. Wound reclosure was photographed at 20x magnification every 2 h for the next 24 h using a spinning disk live–cell imaging system (Zeiss, Jena, Germany). Once the images were recorded, the wound area was measured using the image analysis software Fiji (version 2.1.0/1.53 open-source Fiji/ImageJ) and expressed as a function of time.

### 4.6. ELISA

Secreted FST and activin were quantified in supernatants of HRECs cultured in 30 mM glucose following manufacture instructions.

### 4.7. Statistical Analysis

Unless indicated otherwise, the results are expressed as mean ± SEM. Differences among the groups were evaluated by analysis of variance; the statistical significance of the differences between groups was assessed using the *t*-test when indicated. Significance was defined as *p* < 0.05. Graphs were created using Microsoft Excel 2021 (Microsoft version 16.66.1, Redmond, WA, USA) and images were assembled using Adobe Photoshop (version 23.5.1 20220907, San Jose, CA, USA)

## 5. Conclusions

We conclude that the concentration of VEGF was not the sole determinant of a cell’s response to VEGF. Other agents present in the cell’s microenvironment, such as activin, whose bioactivity was in turn governed by FST, determined the potency of VEGF. The underlying mechanism of activin’s effect involved the increased expression of VE-PTP, which suppressed VEGF-induced signaling. Thus, the consideration of other agents within the microenvironment of the vasculature is a potential strategy to improve current anti-VEGF-based therapies that govern pathologies arising from aberrant vascular homeostasis. 

## Figures and Tables

**Figure 1 ijms-24-08698-f001:**
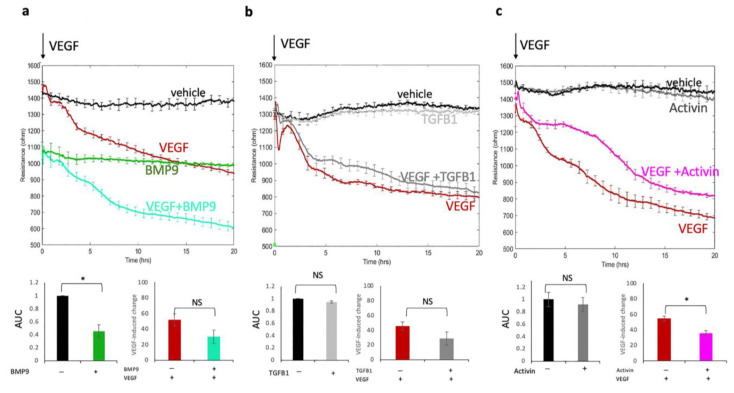
Activin limited VEGF-mediated permeability. Following 48 h pre-treatment with (**a**) BMP-9 (10 ng/mL), (**b**) TGF-β1 (10 ng/mL), or (**c**) activin (50 ng/mL), either 2 nM of VEGF or vehicle was added, and the electrical resistance was monitored for the indicated duration. The bar graphs display the area under the curve (for the entire 20 h duration) and show the effect of each TGF-β family member on permeability in unstimulated (left side) or VEGF-stimulated (right side) cells. While the TEER tracing is of an individual experiment, the bar graph shows the average data for at least three independent experiments. Data are expressed as the means SEM, * *p* < 0.05; NS: not statistically significant, i.e., *p* > 0.05.

**Figure 2 ijms-24-08698-f002:**
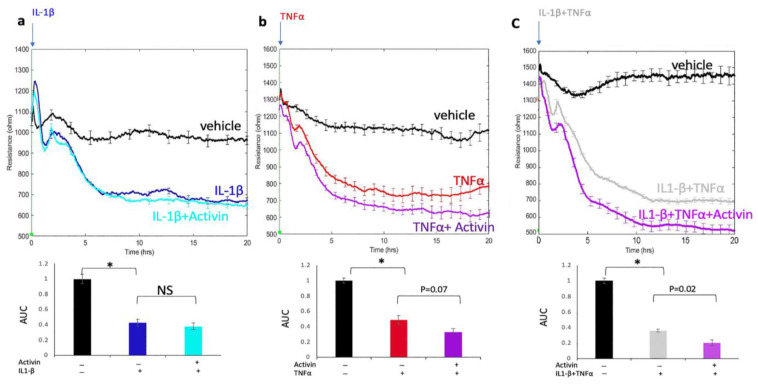
Activin did not attenuate cytokine-induced permeability. HRECs were pre-treated with vehicle or activin, whereupon their responses to (**a**) IL1β (50 ng/mL), (**b**) TNFα (50 ng/mL), or (**c**) both IL1β (50 ng/mL) and TNFα (50 ng/mL) were recorded as in Figure 1. The bar graphs display the area under the curve (AUC) (from 4–20 h) and show the effect of each cytokine on permeability in activin-treated cells. Similar results are observed in at least three independent experiments. Data are expressed as the means SEM for a single representative experiment. NS: not statistically significant, * *p* < 0.05.

**Figure 3 ijms-24-08698-f003:**
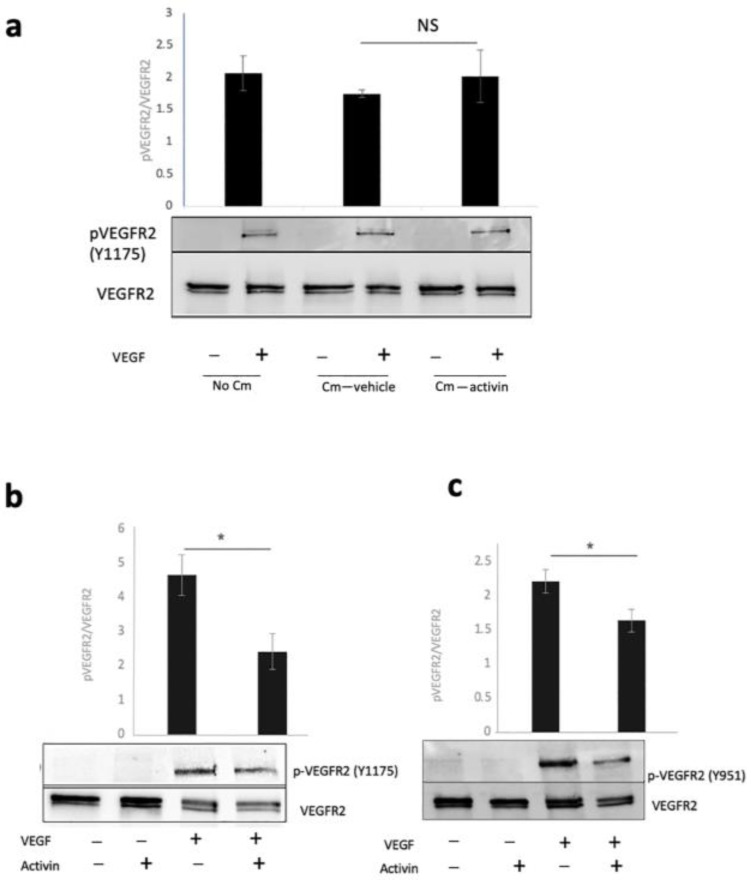
Activin attenuated VEGF-dependent phosphorylation of VEGFR2. (**a**) Confluent HRECs were stimulated for 15 min with PBS (−) or 2 nM VEGF (+) in the presence of no conditioned medium (no CM) or a conditioned medium that was collected from cells that had been pre-incubated for 48 h with vehicle (CM-veh) or activin (CM-activin). The cells were harvested and clarified lysates were subjected to Western blot analysis using the indicated antibodies. The intensity of the indicated bands was quantified and the average pVEGFR2/VEGFR2 ratio from three independent experiments is presented in the bar graph. (**b**,**c**) Confluent HRECs were pre-treated for 48 h with vehicle (−) or activin (+) and then stimulated with PBS (−) or VEGF (+) for 15 min. The cells were harvested and clarified lysates were subjected to Western blot analysis using the indicated antibodies. The intensity of the indicated bands was quantified and the average pVEGFR2/VEGFR2 ratio from three independent experiments is presented in the bar graph. NS: not statistically significant * *p* < 0.05. Uncropped full–length blots are presented in Appendix A.

**Figure 4 ijms-24-08698-f004:**
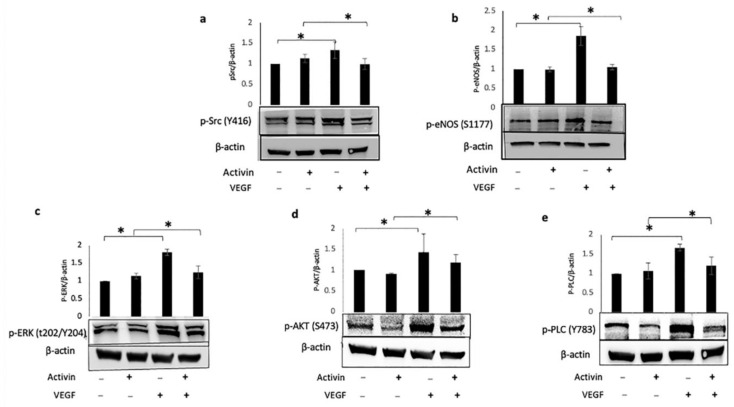
Activin suppressed VEGF-mediated activation of downstream effectors. Confluent HRECs were pre-treated for 48 h with vehicle (−) or activin (+) and then stimulated with PBS (−) or VEGF (+) for 15 min. The cells were harvested and clarified lysates were subjected to Western blot analysis using the indicated antibodies. Bar graphs show the ratio of the amounts of phosphorylated (**a**) Src, (**b**) eNOS, (**c**) ERK1/2, (**d**) AKT, and (**e**) PLCγ. β–actin served as the loading control in three independent experiments. * *p* < 0.05. Uncropped full-length blots are presented in Appendix A.

**Figure 5 ijms-24-08698-f005:**
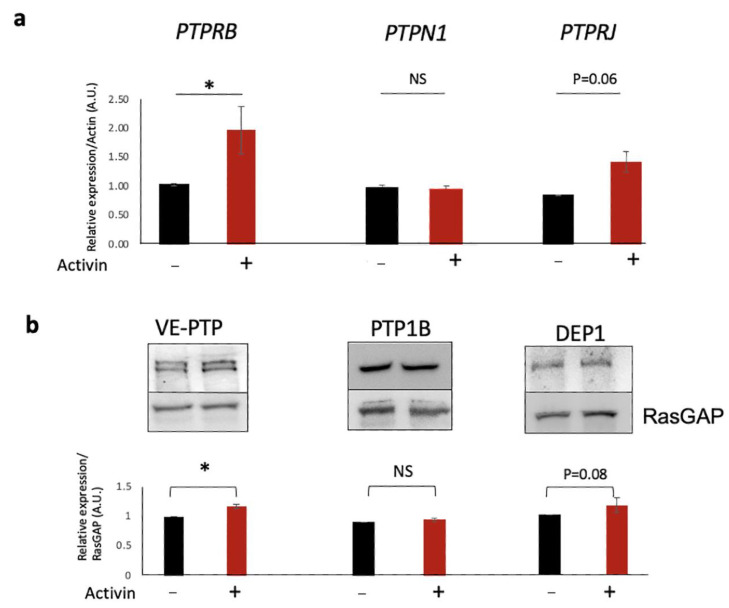
Activin increased the expression of *PTPRB*/VE-PTP. (**a**) Confluent HRECs were treated for 48 h with either vehicle (black bar) or activin (red bar), harvested, and subjected to qRT-PCR analysis. The data in the bar graph are the mean +/−SEM change in expression in response to activin (n = 3–4). At least three independent experiments showed similar results. * *p* < 0.05. (**b**) Cells were treated as described in (**a**), lysed, and subjected to Western blot analysis using the indicated antibodies. The images are of representative Western blots; the bar graphs show the mean +/− SEM change in expression in response to activin in three independent experiments. NS: not statistically significant * *p* < 0.05. Uncropped full-length blots are presented in Appendix A.

**Figure 6 ijms-24-08698-f006:**
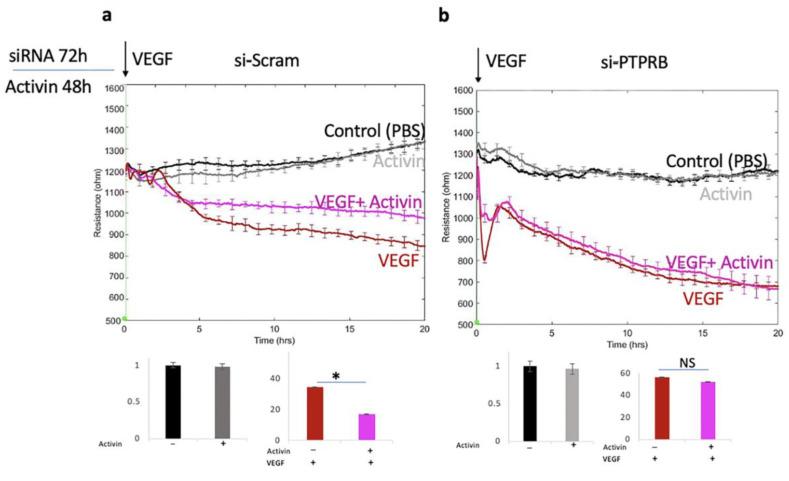
Suppressing the expression of PTPRB eliminated the activin effect. (**a**,**b**) Cells were transfected with the indicated siRNAs, pre-treated with either vehicle or activin, and then stimulated with PBS or 2 nM VEGF. Permeability was monitored as described in Figure 1. The data (area under the curve (AUC) from 0–20 h) was quantified and is presented in the bar graphs; the effects of activin on cells stimulated with PBS and VEGF are on the left and right, respectively. Similar results were observed in three independent experiments. NS: not statistically significant * *p* < 0.05.

**Figure 7 ijms-24-08698-f007:**
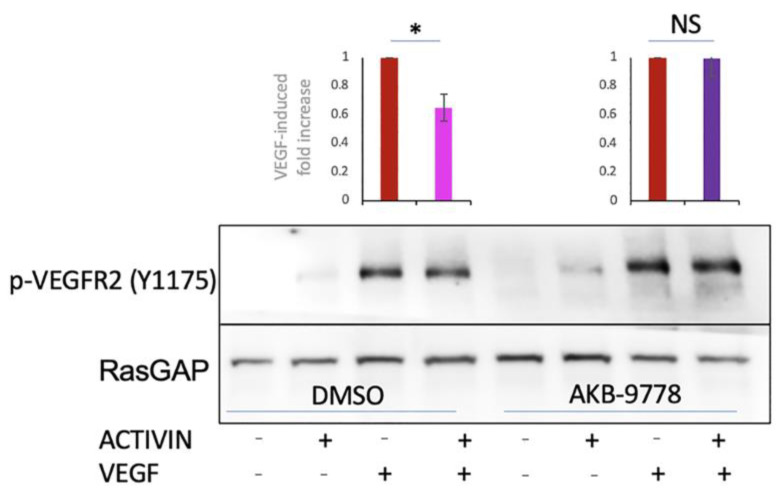
Inhibiting VE-PTP eliminated the activin effect. Confluent HRECs were pre-treated for 48 h with vehicle (−) or activin (+), treated with DMSO or 10 µM AKB-9778 for 30 min, and then stimulated with PBS (−) or VEGF (+) for 15 min. The cells were lysed and clarified lysates were subjected to Western blot analysis using the indicated antibodies. The intensity of the indicated bands was quantified and the average effects of activin on the pVEGFR2/RasGAP ratio from three independent experiments are presented in the bar graph. NS: not statistically significant * *p* < 0.05. Uncropped full–length blots are presented in Appendix A.

**Figure 8 ijms-24-08698-f008:**
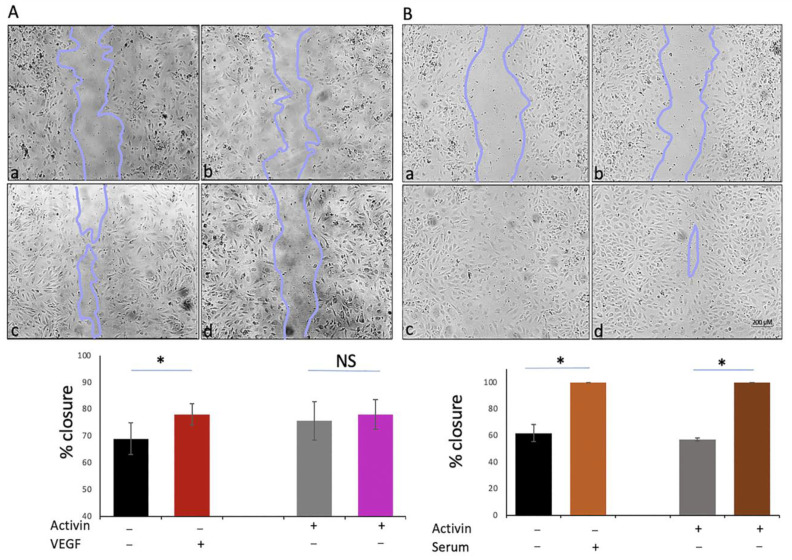
Activin suppressed VEGF-mediated migration. Confluent HRECs were pre-treated for 48 h with vehicle (−) or activin (+) and then subjected to the wound/scratch assay described in the Materials and Methods section. Panels (**A**,**B**) are the response to 2 nM VEGF and 10% serum, respectively. The photographs are of representative images at the 16 h time point; the extent of the closure at this time point was quantified and presented in the bar graphs. (**a**): vehicle, (**b**): activin, (**c**): VEGF or serum, (**d**): activin + VEGF or activin + serum. The data shown in this figure are from a single representative experiment; Scale bars: 200 µM the same results were observed on at least three independent occasions. NS: not statistically significant * *p* ≤ 0.05.

**Figure 9 ijms-24-08698-f009:**
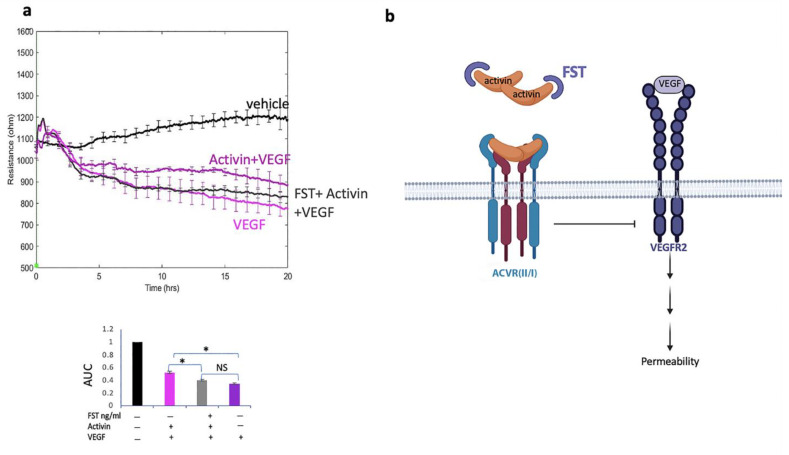
Follistatin overcame the effect of activin. (**a**) Permeability of HRECs was recorded, as in Figure 1, in response to exposure to a vehicle (black), 2 nM VEGF (pink), 50 ng/mL activin + 2 nM VEGF (purple), or activin that had been pre-incubated with FST + VEGF (gray). FST (1000 ng/mL) was pre-incubated with activin A (50 ng/mL) for at least 45 min on ice. The bar graph shows the area under the curve (AUC) from 0–20 h. The data are expressed as a ratio of the AUC in stimulated/stimulated cells. Similar results were observed on at least three independent occasions. (**b**) A diagram illustrating the concept that the relative concentration of all three soluble factors determines the response of cells to VEGF. For instance, activin can suppress VEGF-induced permeability, but only if the level of FST is low. ACVR (II/I): activin receptor type-II and type I. NS: not statistically significant * *p* ≤ 0.05.

**Figure 10 ijms-24-08698-f010:**
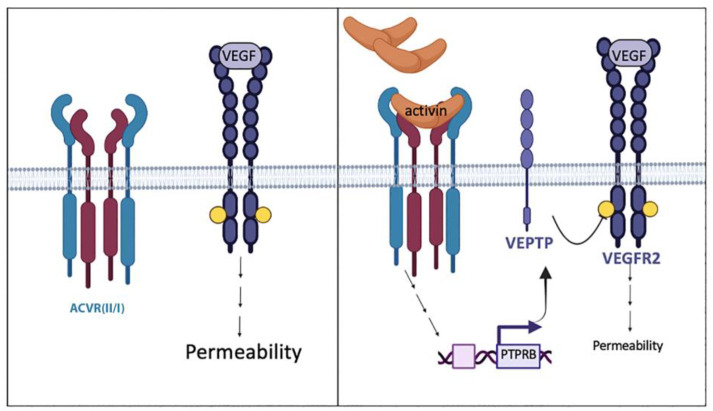
Model illustrating the mechanism by which activin limits VEGF-driven permeability. (**Right**) When VEGF is the only agent present, it activates VEGFR2 and triggers intracellular signaling events that increase permeability. (**Left**) If both activin and VEGF are present, the extent of VEGF-induced permeability is reduced because activin increases the expression of VE-PTP, which attenuates VEGF-mediated activation of VEGFR2.

## Data Availability

The datasets generated during the current study are available from the corresponding author upon reasonable request.

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
