# Peer review of "Activin A Limits VEGF-Induced Permeability via VE-PTP"

_ijms, 2023, doi:10.3390/ijms24108698_

Round 1

Reviewer 1 Report

In geneal, the clinical sucess of neutralizing vascular endothelial growth factor (VEGF) has unequivocally identified VEGF as a driver of retinal edema has been used for several years, and extensively used for other molecualr mechanisms such as dementia, and so on.  It will become a mature techniques and will be needed to further discuss in reuslt and discussion, if possible. 

English language is good, Minor editing of English required.

Author Response

In general, the clinical success of neutralizing vascular endothelial growth factor (VEGF) has unequivocally identified VEGF as a driver of retinal edema has been used for several years, and extensively used for other molecular mechanisms such as dementia, and so on.  It will become a mature technique and will be needed to further discuss in result and discussion, if possible

As requested we added the following paragraph to the end of the Discussion. 

The growing appreciation that dysfunction of the blood brain barrier and increased permeability of the brain vasculature contributes to neurological diseases such as dementia and Alzheimer’s disease [33] suggests that anti-VEGF-based therapies will be used to manage a wider spectrum of disease in the future. 

Reviewer 2 Report

On request of IJMS, I have revised the manuscript titled “Activin A limits VEGF-induced permeability via VE-PTP”, by Basma Baccouche, Lina Lietuvninkas and Andrius Kazlauskas.

The main scope of the present study was verifying the hypothesis that members of the TGF-β family influence VEGF-mediated control of the endothelial cell barrier. To this end, authors compared the effect of BMP-9, TGF-β1, and activin A on VEGF-driven permeability of primary human retinal endothelial cells. According to reported results, while BMP-9 and TGF-β1 had no effect on VEGF-induced permeability, activin A limited the extent to which VEGF relaxed the barrier. Additionally, the authors found that activin A effect is associated with reduced activation of VEGFR2 and its downstream effectors (PLCg, Src, eNOS, Erk 17 and Akt), and with increased expression of vascular endothelial tyrosine phosphatase (VE-PTP).

General comments

Since VEGF is a driver of retinal edema that underlies a variety of blinding conditions, the topic and the contents of the present manuscript are interesting and scientifically relevant. The English language is fine, and the organization of the experimental design is good. Some minor issues need correction to enable the publication of this paper.

Following my opinion and suggestions.

Check carefully all the manuscript to assure that all abbreviations have been specified at their first mention.

Add some other key words.

Line 36. Please, remove “and” before (DR).

Lines 48-51. Please, rephrase the sentence in a clearer way.

Please, in the text use Figure x and not Fig. x.

Figure in the Figure captions not in italics.

Please, use “mL” and not “ml”.

To refer to Supplementary Figures use Figure Sx and not sFig. x. Correct the Supplementary file accordingly.

Line 137. Please, “Conditioned medium experiment” should become an heading.

Line 249. Please correct “suffice”.

Line 364. Heading without the indent. The same in lines 372, 384, 400, 412, 416,

Check all manuscript and remove spaces between numbers of references, and check reference list and correct formatting according to indications of the journal.

Conclusions are missing and must be included.

Author Response

Since VEGF is a driver of retinal edema that underlies a variety of blinding conditions, the topic and the contents of the present manuscript are interesting and scientifically relevant. The English language is fine, and the organization of the experimental design is good. Some minor issues need correction to enable the publication of this paper.

Thank you for this positive evaluation of our manuscript!

Following my opinion and suggestions.

Check carefully all the manuscript to assure that all abbreviations have been specified at their first mention.

As requested, we edited the manuscript to assure that all abbreviations have been specified at their first mention.

Add some other key words.

The following key word have been added: crosstalk, follistatin, VE-PTP

Line 36. Please, remove “and” before (DR).

The requested change has been made.

Lines 48-51. Please, rephrase the sentence in a clearer way:

Crosstalk between VEGF and transforming growth factor β (TGF- β) pathways is essential for proper patterning of the vasculature [7]. While the existence of such crosstalk has been established, its nature, i.e., at the level of the microenvironment and/or intracellular signaling, remains incompletely understood [11].

This section has been rephrased as follows:

Crosstalk between VEGF and transforming growth factor β (TGF- β) pathways is essential for proper patterning of the vasculature [9-10]. Whether such crosstalk occurs at the level of the microenvironment and/or intracellular signaling, remains incompletely understood [11].

Please, in the text use Figure x and not Fig. x.

- The requested change has been made.

Figure in the Figure captions not in italics.

- The requested change has been made.

Please, use “mL” and not “ml”.

- The requested change has been made.

To refer to Supplementary Figures use Figure Sx and not sFig. x. Correct the Supplementary file accordingly.

- The requested changes have been made.

Line 137. Please, “Conditioned medium experiment” should become an heading.

- The requested change has been made.

Line 249. Please correct “suffice”.

- The requested change has been made.

Line 364. Heading without the indent. The same in lines 372, 384, 400, 412, 416,

- The requested changes have been made.

Check all manuscript and remove spaces between numbers of references, and check reference list and correct formatting according to indications of the journal.

- The requested changes have been made.

Conclusions are missing and must be included

The Conclusions have been added; they are pasted below. 

  1. Conclusions

We conclude that the concentration of VEGF was not the sole determinant of a cell’s response to VEGF.  Other agents present in the cell’s microenvironment such as activin, whose bioactivity was in turn governed by FST, determined the potency of VEGF.  The underlying mechanism of activin’s effect involved increased expression of VE-PTP, which suppressed VEGF-induced signaling.  Thus, consideration of other agents within the microenvironment of the vasculature is a potential strategy to improve current anti-VEGF-based therapies to govern pathologies arising from aberrant vascular homeostasis.